New occurrences of fossilized feathers: systematics and taphonomy of the Santana Formation of the Araripe Basin (Cretaceous), NE, Brazil

Prado Gustavo M.E.M. gustavo.marcondes.prado@usp.br gustavo.dino@gmail.com 1 2
Anelli Luiz Eduardo 1
Petri Setembrino 1
Romero Guilherme Raffaeli 1 2
1 Departamento de Geologia Sedimentar e Ambiental, Universidade de São Paulo , São Paulo , Brasil
2 Programa de Pós-Graduação em Gequímica e Geotectônica, Instituto de Geociências, Universidade de São Paulo , São Paulo , Brasil
Young Mark
Electronic publication date: 2016 Jul 7
Publication date: 2016
Volume: 4
Electronic Location ID: e1916
Received 2015 Jun 8; Accepted 2016 Mar 18
Copyright: ©2016 Prado et al.
Copyright year: 2016
Copyright holder: Prado et al.
License: This is an open access article distributed under the terms of the Creative Commons Attribution License, which permits unrestricted use, distribution, reproduction and adaptation in any medium and for any purpose provided that it is properly attributed. For attribution, the original author(s), title, publication source (PeerJ) and either DOI or URL of the article must be cited.
License URL: https://creativecommons.org/licenses/by/4.0/

Keywords: Fossil feathers, Systematic paleontology, Vertebrate paleontology, Araripe Basin, Taphonomy

Funding: Pró-Reitoria de Pesquisa da Universidade de São Paulo Programa de Pós-Graduação em Geoquímica e Geotectônica do Instituto de Geociências da Universidade de São Paulo This project was supported by the Pró-Reitoria de Pesquisa da Universidade de São Paulo, under the undergraduate research program granted to GMEMP in the year 2013 to 2014, and by Programa de Pós-Graduação em Geoquímica e Geotectônica do Instituto de Geociências da Universidade de São Paulo. The funders had no role in study design, data collection and analysis, decision to publish, or preparation of the manuscript.

==============================
Here we describe three fossil feathers from the Early Cretaceous Santana Formation of the Araripe Basin, Brazil. Feathers are the most complex multiform vertebrate integuments; they perform different functions, occurring in both avian and non-avian dinosaurs. Despite their rarity, fossil feathers have been found across the world. Most of the Brazilian feather fossil record comes from the Santana Formation. This formation is composed of two members: Crato (lake) and Romualdo (lagoon); both of which are predominantly reduced deposits, precluding bottom dwelling organisms, resulting in exceptional preservation of the fossils. Despite arid and hot conditions during the Cretaceous, life teemed in the adjacency of this paleolake. Feathered non-avian dinosaurs have not yet been described from the Crato Member, even though there are suggestions of their presence in nearby basins. Our description of the three feathers from the Crato laminated limestone reveals that, despite the small sample size, they can be referred to coelurosaurian theropods. Moreover, based on comparisons with extant feather morphotypes they can be identified as one contour feather and two downy feathers. Despite their rareness and low taxonomic potential, fossilized feathers can offer insights about the paleobiology of its owners and the paleoecology of the Araripe Basin.

Introduction

Feathers are the most complex integuments of vertebrates, due to their variety of forms and roles. This structure is responsible for the thermoregulation, display, protection against radiation, toxicity, buoyancy and even to produce sound (Lucas & Stettenheim, 1972; Dumbacher et al., 2004; Bostiwick & Prum, 2005; Clark, Elias & Prum, 2011; Dimond, Cabin & Brooks, 2011).

Recent molecular studies of feathers suggest a possible phylogenetic hierarchy in the emergence of these elements, connected with the genesis of the tetrapod integuments. During transition from water to land many amphibians shared the same toolkit for coping with environmental changes (e.g., free O2, radiation). On the other hand, reptiles synthesized β-keratin and mammals, α-keratin. Evidence indicates a plausible multiple origin of these structures within Dinosauria (Clarke, 2013), it is possible that the first feathers were present even in the base of the superorder (Xu, 2006; Norell, 2011; Godefroit et al., 2014). Nevertheless, the presence of these elements in basal dinosaurs remains uncertain (Barrett, Evans & Campione, 2015). Despite its earliest origin, it was only in maniraptoriformes that “modern-type feathers” (plumulaceous and pennaceous feathers) have arisen (Xu & Guo, 2009; Clarke, 2013; Godefroit et al., 2013; Han et al., 2014; Koschowitz, Fischer & Sander, 2014).

Regardless, the current idea that feathers may not be a plesiomorphic trait in Dinosauria, and pterosaur pycnofibers may not represent true protofeathers (Barrett, Evans & Campione, 2015), the filaments were reported in a basal ornithischian (Godefroit et al., 2014). Despite being highly contentious, a possible occurrence of filament-feathers in dinosauromorphs (e.g., Lagerpetidae), or even in early members of saurischia clade (e.g., Herrerasaurids), may fills the possible gap between dinosaurs and ornithodirans. However, remains of these animals are often fragmented or unearthed in coarse grain sediments (Nesbitt et al., 2013; Langer et al., 2013; Benton, Forth & Langer, 2014) that preclude the preservation of these fragile structures. However, both filament and true feathers were reported in Jurassic theropods (megalosauroids and coelurosaurs), suggesting that this character may be present early in theropods (Rauhut et al., 2012; Foth, Tischlinger & Rauhut, 2014; Xu et al., 2014). Despite the broad distribution within this clade, “true feathers” undoubtedly made of β-keratin, only started to be synthesized later, in derived theropod dinosaurs, such as coelurosaurs (Prum & Brush, 2002; Fucheng, Zhou & Dyke, 2006; Xu, 2006; Xu & Guo, 2009; Norell, 2011; Clarke, 2013).

Feathers were always thought to be associated with flight (Feduccia, 1993; Martin, 1998). Recent authors (Dial, 2003; Dial, Jackson & Segre, 2008; Heers & Dial, 2012) demonstrated a disconnection between feather and flight. In addition, evidence in non-avian dinosaurs, such as the dromaeosaurids (Han et al., 2014), oviraptorosaurians (Qiang et al., 1998; Xu, Zheng & You, 2010), ornithomimids (Zelenitsky et al., 2012), or tyrannosauroids (Xu et al., 2012) makes this hypothesis even more unlikely. Other controversial interpretations consider that feathers originated to perform the thermoregulation functions (D’Emic, 2015; Myhrvold, 2015).

Recent studies indicate that dinosaurs were possibly mesothermic, suggesting no relation with the advent of homeothermy (Grady et al., 2014; Grady et al., 2015). Furthermore, the endothermy may have originated only during or briefly after the advent of flight, since this activity require high metabolism, with high consumption of O2 and a low accumulation of C3H6O3 (lactic acid). Whether analogous or homologous, thermoregulation control may have evolved with the help of integumentary coverings (pycnofibers and feathers), and these elements are suggestive of an ornithodiran wide trait, even though this is highly polemic (Unwin & Bakhurina, 1994; Unwin, 1998; Ruben & Jones, 2000).

A tactile function as the possible cause of the origin of these elements was recently proposed by Persons & Currie (2015). According to these authors, beyond its hygienic roles, this hypothesis can satisfactorily explain the origin of filamentous-type integuments that could be located in the face of its owners for semi-fossorial habits. Because the preservation of these elements is unusual in the fossil record, their proposal must require further evidence.

Another reason for the rise of feathers may be the ability to maintain a social relationship between individuals, as extant birds currently do today. It also explains the evolution of the morphotypes, and the wide range of color patterns which have been arisen in avian-dinosaurs, once the sexual selection could be the main driver for their evolution (Dimond, Cabin & Brooks, 2011; Koschowitz, Fischer & Sander, 2014).

Because feathers are very delicate features, they rarely survive the physicochemical process that follow their burial. Thus, they are usually found as: (i) carbonized and iron traces, (ii) inclusions in ambers and coprolites, (iii) and as imprints (Wetmore, 1943; Martins-Neto & Kellner, 1988; Davis & Briggs, 1995; Vinther et al., 2008; McKellar et al., 2011; Vitek et al., 2013).

Only a few deposits possess vestiges of feathers, not exceeding 50 around the world (Kellner, 2002). Despite their rareness, there is a relatively cosmopolitan distribution of these structures, extending from the Middle Jurassic to the Neogene. Feathers were found in Mesozoic and Cenozoic deposits and ambers, in Europe and North America (Kellner, 2002; McKellar et al., 2011; Valentin et al., 2014). On the Southern Hemisphere, they were found in Australia (Talent, Duncan & Hanby, 1966; Waldman, 1970), and South America (Kellner, 2002; Sayão & Uejima, 2009; Clarke et al., 2010; Leite & Hessel, 2011; Sayão, Saraiva & Uejima, 2011; Mansilla et al., 2013; Carvalho et al., 2015a; Carvalho et al., 2015b).

The first occurrence of fossil feathers in Brazil was reported from the Oligocene shales of the Tremembé Formation, Taubaté Basin (Shufeldt, 1916), followed by discoveries in other two geologic units: the Aptian-Albian limestones of the Santana Formation of the Araripe Basin (Kellner, 2002; Sayão & Uejima, 2009; Leite & Hessel, 2011; Sayão, Saraiva & Uejima, 2011) and Miocene limestones of the Pirabas Formation of the Barreirinhas Basin discovered by Ackerman (1964). While the Pirabas Formation exhibited a single occurrence of feathers since 1964, both Santana and Tremembé formations are responsible for the major records of this type of fossil (Kellner, 2002) (Table 1).

Table 1 Brazilian feather occurrences.

The Brazilian fossil record of feathers formally described.

Feather	Deposit	Age	Preservation	Observations	Reference	
One primary remex	Tremembé Fm	Paleogene (Oligocene)	Carbonized	First record in the country	Shufeldt (1916)	
One contour feather	Tremembé Fm	Paleogene (Oligocene)	Carbonized	Feather assigned to a Turdidae (Turdus rufiventris)	Santos (1950)	
Two pennaceous feathers	Pirabas Fm	Neogene (Miocene)	Carbonized	Possible semiplumes or contour feathers	Ackerman (1964)	
One primary remex	Santana Fm	Cretaceous (Aptian/Albian)	Limonitc/ Imprint	Asymmetrical feather attributed to birds	Martins-neto & Kellner (1988)	
Contour feathers	Tremembé Fm	Paleogene (Oligocene)	Carbonized/ Imprint	Several feathers associated with skeleton of Taubacrex granivora	Alvarenga (1988)	
Semiplume	Santana Fm	Cretaceous (Aptian/Albian)	Carbonized	Feather assigned to passerine birds	Martill & Filgueira (1994)	
Down feather	Santana Fm	Cretaceous (Aptian/Albian)	Carbonized	Feather assigned to thermoregulation function of a bird	Kellner, Maisey & Campos (1994)	
Contour feather	Santana Fm	Cretaceous (Aptian/Albian)	Melanosome preservation	Feather with (banded) color pattern preserved.	Martill & Frey (1995)	
One symmetrical feather	Santana Fm	Cretaceous (Aptian/Albian)	Carbonized	The biggest isolated feather associated with ectoparasite eggs. Assigned to a bird.	Martill & Davis (1998); Martill & Davis (2001)	
Plumulaceous feathers	Santana Fm	Cretaceous (Aptian/Albian)	No data. Presumably carbonized	One plume and one semiplume	Sayão & Uejima (2009)	
Plumulaceous feathers	Santana Fm	Cretaceous (Aptian/Albian)	Carbonized	Eight contour feathers	Leite & Hessel (2011)	
Down feather	Santana Fm	Cretaceous (Aptian/Albian)	Carbonized	Feathers assigned to a bird	Sayão, Saraiva & Uejima (2011)	
Several rectrices, remiges and filamentous feathers (possibly contour feathers)	Santana Fm	Cretaceous (Aptian/Albian)	Carbonized	Several feathers associated with a skeleton of the enantiornithe Cratoavis cearensis. First formally description of a Mesozoic bird in Brazil.	Carvalho et al. (2015a) and Carvalho et al. (2015b)	

The fauna of Crato and Romualdo members are probably allochtonous. The diversified biota was seemingly laid down in nearby paleolake shorelines (Naish, Martill & Frey, 2004). A swift deposition in Crato must be responsible for the good preservation of the fossils. However, the presence of the vertebrate remains in Romualdo is often explained by the “drifting hypothesis” proposed by Naish, Martill & Frey (2004): Carcasses might have been transported by rivers for up to tens of kilometers before reaching the calm waters of the Romualdo lagoon. Once there, they were preserved by the process of ‘encapsulation’, also known as “The Medusa effect” (Martill, 1989). The Romualdo was characterized by the maximum marine transgression, and salty waters entered this basin in an N-NW direction (Assine, 1994). Therefore, the carcasses of these animals may have been dragged in at water. The drifting hypothesis is also able to explain the preservation of the isolated and often disarticulated bones of pterosaurs, dinosaurs and other aerial and terrestrial vertebrates. Nonetheless, the absence of ichnofossils in the rocks of this unit requires further investigations. Although highly contentious, two other hypotheses may explain the presence of terrestrial vertebrates in the Araripe Basin. Mass mortality events caused by environmental changes (e.g., chemoclinal alterations, remobilization of the anoxic layers) could expose remains of aquatic animals on the shores of the paleolake, attracting animals in order to prey, where they may have become stuck in the soft and deep sediments (Olson & Alvarenga, 2002; Varricchio et al., 2008). The third hypothesis is based on bacteria Clostridium botulinum, responsible for mass mortality of aquatic birds (Duncan & Jensen, 1976; Varricchio, 1995). However, these events are restricted to fishes, requiring further evidences of this phenomenon, such as high bone concentration of different vertebrates in the same strata (Varricchio, 1995; Martill, 1997; Martill, Brito & Washington-Evans, 2008).

On this paper, we report three new occurrences of fossil feathers, from Cretaceous of Crato Member of the Araripe Basin and propose a systematic approach to these fossils, according to the available data. Preliminary discussions about the taphonomy and paleoecology are presented; the presence of avian dinosaurs and their paleoecology are also discussed.

Figure 1 Araripe Basin locality and lithology.

The Araripe Basin locality, the stratigraphic columns, units and chronology. (Adapted and modified from Coimbra, Arai & Carreño, 2002; Vianna & Neumann, 2002; Assine, 2007.)

Geologic setting

The Araripe Basin (Fig. 1) is located in the northeastern Brazil between longitude 38°30′W to 40°50′W, and latitude 7°.05′S to 7°50′S (Coimbra, Arai & Carreño, 2002; Vianna & Neumann, 2002), and extends approximately 5.500 to 8.000 Km2 across three states (Ceará, Pernambuco and Piauí). The exceptional fossil preservation of the Crato Member was highlighted by Martill, Bechly & Loveridge (2007), who placed it in a Konservat-Lagerstätte. The geology of this basin has been studied since the 19th century (Carvalho & Santos, 2005), with differemt interpretations (Maisey, 1991; Assine, 1992; Martill, 1993; Vianna & Neumann, 2002; Carvalho & Santos, 2005; Assine, 2007; Martill, Bechly & Loveridge, 2007).

Assine (1992), Assine (1994) and Assine (2007) surveyed in detail the entire basin, establishing a stratigraphic subdivision, based on the recommendations of the Brazilian Code of Stratigraphy Nomenclature. This classification is herein followed. The Santana Formation is subdivided into the Crato Member (base), and the Romualdo Member. These units have different lithologies that reflect their distinct depositional environments. Many of the exquisitely preserved fossils of the Araripe Basin come from the Crato Member strata, which is characterized by micritic laminated limestones intercalated with shales and mudstones of varied thicknesses. The unit was formed in a lacustrine environment with brackish water of dubious depth, and reducing conditions in the bottom (Assine, 1992; Assine, 1994; Assine, 2007; Martill, Bechly & Loveridge, 2007; Heimhofer et al., 2010).

Since the studied specimens were the product of apprehension (illegal fossil trade), it was not possible to get their stratigraphic positions. However, the laminated limestones (LL) of the Crato Member, are well known worldwide by geologists and paleontologists. Since the LL only occur in this unit, it was possible to assign these fossils to this specific layer.

Figure 2 Samples (Feathers and fish).

Fossilized feathers and fish of the Santana Formation. (A) GP/2E-8771; (B) GP/2E-7854; (C) GP/2E-7853; (D) Detail of the barbs and barbules of GP/2E-8771; (E) Detail of the barbs and barbules of GP/2E-7854. Arrows indicate the barbules. (F) Photograph of the umbilicus proximallis; (G) Interpretative drawing of the calamus; (H–I) The GP/2E-7853 specimen; (I) Detail of the Dastilbe sp. fossil fish. Legend of (G): CL, Calamus; BI, Isolated Barb; VX, Vexillum (vanes); RQ, Rachis. Scale bars: (A, C, F–G) 2 mm; (B) 5 mm; (C) 2 mm; (D, H–I) 10 mm; (E) 1 mm (F) Detail: 2, 6 mm.

Santana Formation ostracods and palynomorphs were studied by Coimbra, Arai & Carreño (2002), but only the palynomorphs were fitted for biostratigraphy correlations with nearby basins. Crato is Aptian in age (∼120 Ma) and Romualdo is Albian (∼111 Ma).

Materials and Methods

Three specimens were studied and described, following the terminology of Lucas & Stettenheim (1972), Sick (1984) and Proctor & Lynch (1993). These fossils were apprehended by the Brazilian Federal Police and the IPHAN (Institute of National Historical and Artistic Heritage) and are deposited in the Paleontological Collection of the Laboratory of Systematic Paleontology from the Institute of Geosciences, of the University of São Paulo, in the city of São Paulo. The specimens received the registered numbers: GP/2E-7853, GP/2E-7854 and GP/2E-8771. The acronyms used in the collection assign the “GP” to Geology and Paleontology sets, and ‘2E’, to the vertebrate set.

All specimens were photographed using a millimeter-scale stand with Canon EOS REBEL T3 with aperture of 100 mm and under a stereomicroscope Carl Zeiss with a capture system AxioCam ICC3 and using the AxioVision LE software. The specimens were measured with a caliper and the AxioVision LE software. Specific portions of the feathers, such as barbs and rachis were measured. The difference between every portion (i.e., calamus, larger barb, minor barb and rachis) was compared with the total size of the length. These measurements were used to infer the morphology and to classify them according to the terminology of extant feathers (Lucas & Stettenheim, 1972; Sick, 1984; Proctor & Lynch, 1993).

Results

Systematic palaeontology

Order Saurischia Seeley, 1888	
Suborder Theropoda Marsh, 1881	
Division Coelurosauria Von Huene1914 sensu Gauthier, 1986	
Family Incertae sedis	
(Figs. 2B–2C)	

Material: GP/2E-7853 (Fig. 2C).

Horizon: Crato Member, Santana Formation, Araripe Basin.

Lithology: Weathered (beige) micritic laminated limestone.

Age: Lower Cretaceous (Aptian).

Description: This specimen is a complete feather with reduced dimensions compared to other morphotypes (i.e., contour feathers and pennaceous feathers) where it is possible to see the sizes and differences (Tables 2 and 3). It presents an orange coloration. Barbules are not clearly visible and are presented only in some regions of the barbs. The rachis consists of a thin line. The distal extremity presents ramifications, where barbs with different length originate. As well as in other feathers, the calamus was not preserved. It may represent the morphotype “IIIb” of Prum & Brush (2002) evolutionary model, and by its morphology it can be associated to plumulaceous feathers where rachises are generally thin and barbs are open vaned.

Table 2 Measures of the new specimens.

Values of the measures of the three specimens.

Specimen	Width	Length	Larger barb	Minor barb	Calamus	Rachis	Rachis thickness	
GP/2E-7853	12,36	16,14	8,65	4,85	NP	9,43	0,49	
GP/2E-7854	12,76	19,00	17,83	4,30	0,24	12,03	0,49	
GP/2E-8771	15,63	33,50	16,45	4,12	NP	29,35	0,03	
Notes.

NP Not present

Dimensions are in mm.

Table 3 Feather portions calculation.

Difference in percentage between portions of the feathers compared to the maximum length.

Structure	Percentage	
	GP/2E-7853	GP/2E-7854	GP/2E-8771	
Larger barb	46,41	6,16	50,90	
Minor barb	69,95	77,37	87,70	
Calamus	ND	1,26	ND	
Rachis	41,57	36,68	12,39	
Notes.

ND No data available

Measures: See Table 2, first row. Dimensions are in mm.

Taphonomy: The color of this specimen (orange/reddish), suggests that the fossil may be preserved as an iron oxide. The matrix light beige color may be the result of slight weathering, calcified filaments and dendritic crystals of sphalerite (Martill & Davis, 2001; Heimhofer et al., 2010).

Diagnosis: Despite a fairly generic morphotype, this specimen has typical plumulaceous feather morphology due to the presence of very well delineated rachis and barbs of varying sizes. The rachis is 8.27% longer than the larger barb. Since GP/2E-7853 has a longer rachis than the largest barbs, and fluffy aspect, dimension, and morphology, this feather is assigned as a downy feather (Lucas & Stettenheim, 1972; Sick, 1984; Proctor & Lynch, 1993). It is not possible to observe the presence of the calamus. Generally, because of their fragility and small size (in life it may represent only 1.5% of the total length of the feather), this portion, commonly, does not preserve in the fossil record (Lucas & Stettenheim, 1972; Kellner, 2002). In the sample matrix, a nearly complete skeleton of a small fish is associated (Figs. 2H–2I), classified as Dastilbe sp. (Maisey, 1991; Dietze, 2007; Martill, Bechly & Loveridge, 2007)

Family Incertae sedis	

Material: GP/2E-7854 (Fig. 2B).

Horizon: Crato Member, Santana Formation, Araripe Basin.

Lithology: Weathered (beige) micritic laminated limestone.

Age: Lower Cretaceous (Aptian).

Description: The proximal portion is degraded, though, the rachises are visible. Several barbs with different length originate from them. It is also possible to notice the presence of vestigial barbules (Fig. 2E). The calamus is a slight line, and together with GP/2E-7853, this feather can also be assigned to “IIIb morphotype (Prum & Brush, 2002).

Measures: See Table 2, second row. Dimensions in mm.

Taphonomy: Similar to GP/2E-7853, this sample is a small feather, but complete. The color varies from between the proximal to distal portion of the vanes, among lighter to darker brownish tones, as a consequence of different preservation in carbonaceous traces (Davis & Briggs, 1995).

Diagnosis: This specimen also presents the typical morphology of the plumulaceous feathers, classified as downy feathers. On the umbilicus proximallis portion (Figs. 2F–2G), the slight line structure consisted of an external molt that is interpreted as the vestige of the calamus. By the preservational characteristics (e.g., external mould, lack of organic remains), the evidence suggests that this portion was degraded during the taphocenosis, or geochemical processes that followed the burial (diagenesis). In GP/2E-7854 the difference between the rachis and the longest barbs of GP/2E-7854 is 48.21%.

Order Saurischia Seeley, 1888	
Suborder Therapoda Marsh, 1881	
Division Coelurosauria Von Huene1914 sensu Gauthier, 1986	
Subdivision Maniraptoriformes Holtz, 1996	
Family Incertae sedis	
(Fig. 2A)	

Material: GP/2E-8771 (Fig. 2A).

Horizon: Crato Member, Santana Formation, Araripe Basin.

Lithology: Grayish micritic laminated limestone.

Age: Lower Cretaceous (Aptian).

Description: This specimen is a complete feather and the largest of the three, compared with the two previously described (Tables 2 and 3). Different barbs with variable lengths originate from a slight rachis. The barbules are clearly visible (Fig. 2D), and vary in size. In extant feathers, vanes are united by the ‘hooklets’ (structures similar to hooks) (Lucas & Stettenheim, 1972; Sick, 1984), but hooklets are not preserved.

Measures: See Table 2, third row. Dimensions are in mm.

Taphonomy: This specimen, like GP/2E-7853 and GP/2E-7854, also occurs in a limestone matrix.

Due to the blackish color of the fossil, this feather possible was preserved as carbonized trace, since it is the common type of preservation (Davis & Briggs, 1995).

Diagnosis: According to morphology, GP/2E-8771 is associated to the typical extant contour feathers or semiplumes. Attached to the basal part (the umbilicus), structure reminding afterfeather emerged, forming a V shape, larger than the vanes (Lucas & Stettenheim, 1972). However it does not exhibit afterfeather diagnostic features, like “slight rachis” or umbilical origin. The barbules are present suggesting some degree of cohesion between barbs. However no ‘hooklets’ (barbicels) are preserved on this specimen. The characteristic that distinguish this specimen from the other two previously described is the hue color of the matrix. This feature is suggestive that this sediment was not exposed to weathering processes which usually change the rock color (Martill & Frey, 1995). In an attempt to turn the fossils more attractive, some portions of the feather were degraded with a scraper tool by the illegal dealers, especially on the portion where the calamus was supposed to be found. The GP/2E-8771 is the only Mesozoic feather described here that could be assigned to the crown group Aves, since all of its characteristics are very similar to modern morphotypes. However, because this morphotype were also found in non-avian dinosaurs, a parsimonious assignment is that it belonged to the maniraptoran clade. The specimens possess a morphotype similar to semiplumes, with an apparent aftershaft on the proximallis portion. However, this structure may not represent the semiplume. The rachis is 43.95% shorter than the longest barb. The morphology of this feather is similar to the type of afterfeathers that possess a long, narrow rachis with shorter vanes. The hyporachis is almost the same length of the afterfeather. In extant cases, these feathers are related to birds of Galliformes, Tinamiformes and Trogoniformes orders (Lucas & Stettenheim, 1972).

Discussion

Isolated feathers have been described formally in many papers (Kellner, 2002), however, different from fish scales, mollusks shells, plant trunks and leaves, none of them, to date, have received a proper taxonomic treatment. The main reason for the lack of systematic (taxonomic) procedure may be caused by their scarcity and common occurrence as isolated feathers which hamper taxonomic assignment. However, it does not prevent other systematic works from being performed. Despite the taphonomic significance, this perfunctory treatment can also be an issue that systematists simply ignore, once these elements demonstrate low taxonomic interest (i.e., low potential to assign a new taxon). Nevertheless, the characteristics of a feather also allow its recognition as part of a family or subfamily. Rautian (1978) applied a different taxonomic approach to these elements, once their existence represented (at that time) a diagnostic feature of a new bird taxon. Nowadays, this method proves to be problematic, since non-avian dinosaurs also possessed them, demanding a different way to assess their taxonomic value. Williamson et al. (2009) applied a systematic procedure, which is very similar to the present paper, to describe feathers from the Upper Cretaceous of New Mexico. However, their approach was superficial and brief, attending only to geological features without any other information such as taphonomy and paleobiology. Their description failed to explain and support the taxonomic assignment attributed by these authors. The approach we propose is a simple and parsimonious approach to describe fossilized feathers, assigning their morphotypes to the basal animals that possessed them according to the fossil record of non-avian and avian dinosaurs.

Based on the morphology (barbs that originates from a scanty rachis; absence of barbules; small dimension between morphotypes; bigger length of the barb than the rachis; and, fluffy aspect), and comparison of the specific portions (Table 3) of the three feathers, it was possible to classify these feathers to plumulaceous and pennaceous morphotypes (Table 4). The occurrence of these morphotypes are wide in the extant class Aves, once they are present beyond the semiplumes and are located in the apterium portions of most birds (Lucas & Stettenheim, 1972). According to the fossil record, these structures could also belong to non-avian dinosaurs, making the taxonomic assignment even harder to be inferred (Prum & Brush, 2002; Fucheng, Zhou & Dyke, 2006; Xu & Guo, 2009). Moreover, since all specimens are from the Mesozoic (period marked by “evolutionary experiments”), at least two specimens (GP/2E-7853 and GP/2E7854) deserved more attention by their generic morphotypes, which resemble, ontogenetically, early and evolutionarily basal feathers. Despite the controversy over morphotype diversity provided by the squeeze effect diagenesis (Foth, 2012) and, with the apparent decrease of species suggested by the fossil record (Fucheng, Zhou & Dyke, 2006; Xu & Guo, 2009; Xu, Zheng & You, 2010), it was possible to associate both specimens aforementioned to their evolutionary stages, as it was proposed by the literature (Prum & Brush, 2002; Xu & Guo, 2009).

Table 4 Taxonomic assignment.

Classification of the described feathers.

Specimen	Morphotype	Evolutionary-developmental model	Morphotype model present of the fossil record	
GP/2E-7853	Downy feathers	IIIb	Morphotype 4	
GP/2E-7854	
GP/2E-8771	Semiplume (Contour feather)	IIIa+b	Morphotype 6	

The preservation of the macro-structures and identification of morphotype and size, allow suggestions into the possible roles of feathers during life, their placement throughout their body, and proportions of the owners (Lucas & Stettenheim, 1972; Sick, 1984; Proctor & Lynch, 1993). According to Lucas & Stettenheim (1972), both feathers have the morphotype, size (length between 2.5 and 17 mm) and general aspects similar to auricular feathers. Regardless of the ontogenetic possibilities (Xu, Zheng & You, 2010; Zelenitsky et al., 2012), which are difficult to be inferred by isolated feathers, the parsimonious explanation is that they represent adult forms. If this identification is correct, the animal that possessed these elements may not have had large dimensions, i.e., not exceeding the size of a domesticated chicken (Lucas & Stettenheim, 1972). The fact that a small sized euenantiornithe was found in this deposit, as well as many other small isolated feathers (the larger measured, is 85 ×11 mm) also corroborate this idea (Kellner, 2002; Sayão, Saraiva & Uejima, 2011; Leite & Hessel, 2011). Thus, it must have a similar role to the extant birds, where the main function is in ear protection (Lucas & Stettenheim, 1972). The other feather (GP/2E-8771), a contour feather, was suggested to have also taken the same protective function. However, it might also have functioned in thermoregulation. Nevertheless, even in basal coelurosaurs, they may have had other roles such as display, shielding nests, etc. (Turner, Makovicky & Norell, 2007). Other lines of evidence suggest that dinosaurs already possessed visual acuity, with nocturnal or crepuscular behavior, and abilities to visually communicate might have been present in the Mesozoic (Varricchio, Martin & Katsura, 2007; Xu, Zheng & You, 2009; Schmitz & Motani, 2011; Koschowitz, Fischer & Sander, 2014).

The morphotype GP/2E-8771, and the position throughout the body, point to the possibility that this feather might have favored camouflage and communication between its owners, as seen in modern birds (Gluckman & Cardoso, 2010). In addition, based in extant examples, it could also have assumed a sexual role, similar to extant birds with iridescent and colorful feathers, such as peacocks (Zi et al., 2003) and birds-of-paradise (Irestedt et al., 2009). However, this interpretation is merely speculative, since the true colour and position on the body is uncertain.

Sedimentary deposition, paleoenvironment, and taphonomy

The Santana Formation, during the Aptian-Albian, was under two different depositional systems. The Crato deposits were laid down under a restricted lacustrine brackish water environment. The Romualdo deposits, on the other hand, were thought to be formed under a lagoon in seasonal contact with marine waters, or at moments of marine regression-transgressions (Assine, 1994; Assine, 2007; Martill, Bechly & Loveridge, 2007). An unconformity separates these units; stratas of shales and evaporites that characterizes the ‘Ipubi Layers’ occur with varied thickness and lateral continuum, suggesting the possible shallowing of the water column (Assine, 2007; Martill, Loveridge & Heimhofer, 2007).

According to paleontological and sedimentary evidence, such as palynomorphs and evaporites, the Crato Member was laid down under clear and relatively shallow waters during an arid and dry climate, where the carbonate sediments were deposited in a low energetic input with formation of halite and anhydrite minerals (Assine, 1994; Silva et al., 2003; Assine, 2007; Martill, Loveridge & Heimhofer, 2007). As suggested elsewhere (Martill, Bechly & Loveridge, 2007), this anoxic and hypersaline environment prevented the presence of the bottom-dwelling organisms, so the salinity content might have been higher than the osmotic toleration (Martill, 1993; Martill, Bechly & Loveridge, 2007; Martill, Loveridge & Heimhofer, 2007; Martill, Loveridge & Heimhofer, 2008).

The source of Crato calcareous deposits might be stromatolites from the border of the basin (Martill, 1993; Srivastava, 1996), but algal bloom events might also have occurred (Martill, Bechly & Loveridge, 2007; Martill, Loveridge & Heimhofer, 2008). The presence of articulated, undisturbed fossils, and pseudomorphs of pyrite and marcasite, indicate that the reducing conditions prevailed at the bottom of the paleolake, enabling the exquisite preservation of non-resistant tissues (Fielding, Martill & Naish, 2005; Martill, Bechly & Loveridge, 2007; Pinheiro et al., 2012; Simões, Caldwell & Kellner, 2014; Barling et al., 2015). The high degree of articulation and the exquisite preservation suggest a low energy environment, without any or significant carcass transportation, as well as disturbance by scavenging organisms (Fielding, Martill & Naish, 2005; Martill, Bechly & Loveridge, 2007; Báez, Moura & Gómez, 2009; Figueiredo & Kellner, 2009; Pinheiro et al., 2012; Barling et al., 2015).

Despite this “harsh” environment, this unit is remarkable in the abundant biota, preserved with a high degree of fidelity. The vertebrate fauna is composed primarily by crocodiles, turtles, frogs, birds, pterosaurs, and numerous fishes (Maisey, 1991; Martill, 1993; Martill, 1997; Fielding, Martill & Naish, 2005; Martill, Bechly & Loveridge, 2007; Martill, Brito & Washington-Evans, 2008; Figueiredo & Kellner, 2009; Pinheiro et al., 2012; Simões, Caldwell & Kellner, 2014; Oliveira & Kellner, 2015). Invertebrate animals were also abundant, with the mainly occurrences of arthropods and mollusks (Maisey, 1991; Martill, 1993; Grimaldi & Engel, 2005; Martill, Bechly & Loveridge, 2007; Barling et al., 2015). The flora was also exuberant and diversified (Martill, Bechly & Loveridge, 2007; Martill et al., 2012; Mohr et al., 2015), characterized by macro and microfossils pteridophytes, gymnosperms, angiosperms, palynomorphs, pollen, seeds, etc. (cf. Maisey, 1991; Martill, 1993; Martill, Bechly & Loveridge, 2007; Martill et al., 2012). The fauna of the Crato Member may have been autochthonous (Naish, Martill & Frey, 2004), however, the Santana Formation terrestrial vertebrates thrived at different geographical regions through time, indicated by evidence in other adjacent basins (Carvalho & Gonçalves, 1994; Carvalho, 1995; Carvalho & Araújo, 1995; Carvalho, Viana & Filho, 1995; Carvalho & Pedrão, 1998). Nevertheless, in both lagerstätten units (Crato and Romualdo), animals were well adapted to the arid and dry climate (Naish, Martill & Frey, 2004; Martill, Bechly & Loveridge, 2007; Heimhofer et al., 2010). Many of the animals may have lived in the surroundings of the palaeolake allowing a high diversity of plants, especially angiosperms. At least and especially in the Crato Member, it is possible that birds lived by the shore which enhanced the probability of these elements being preserved, and because of the absence of non-avian dinosaur bones, it is possible that these animals lived inland, off the palaeolake surroundings. This niche occupation pattern would explain the large record of fossil feathers recovered from this unit.

From the three specimens studied, only GP/2E-7853 shows coloration (reddish/orange) that is typical of the iron oxides-hydroxides, possibly limonite. This type of preservation was also observed in other feathers from the same provenance (Maisey, 1991; Martins-Neto & Kellner, 1988; Martill & Frey, 1995; Martill & Davis, 2001). The remaining specimens may be preserved as incarbonization, one of the most common type of preservation of organic molecules, with characteristic dark black hue (Tegelaar et al., 1989; Davis & Briggs, 1995; Kellner, 2002; Briggs, 2003). However, to be sure of this chemical composition further investigations are needed. The process of preservation also explains the absence of hooklets in all specimens, since these structures are very delicate, their presence is not expected, so this feature is not common in preserved feathers in rocks being only present in feathers within amber (Davis & Briggs, 1995; Laybourne, Deedrick & Hueber, 1994; Perrichot et al., 2008; McKellar et al., 2011; Thomas et al., 2014).

The main hypothesis that explains the presence of isolated feathers in the fossil record, as is in the Crato, assumes that these elements may have been blown into the paleolake by strong winds events. Once they have reached the lake, these feathers would sink quickly, reaching the bottom in seconds to a few minutes, where they might be rapidly buried (Martill & Davis, 2001).

Birds normally lose their feathers during ontogenetic phases or seasonably. It may also happen under stress conditions when living birds have a tendency to release rectrices and semiplumes (Sick, 1984). Possibly it happens also as the result of predation by aquatic predators, however there is no evidence of this in Crato Member (Davenport, 1979; French, 1981; Perry et al., 2013; O’Brien et al., 2014). Despite the possibility that birds were also prey, the fossil record of the established trophic chain do not yet show that these animals were a food source for other organisms. Furthermore, coprolites did not yet provide evidence of this diet (Maisey, 1991; Martill, 1993; Lima et al., 2007).

The carbonate concretions of the Romualdo Member of the Santana Formation, provided a record of at least four non-avian theropods, with the possibility of a fifth (Machado & Kellner, 2007), interpreted as a rib of an unknown theropod. Only theropods were found in this unit so the dinosaur fauna of the Araripe Basin consists of two spinosaurids, Irritator challengeri (Martill et al., 1996) and its possible synonym, Angaturama limai (Kellner & Campos, 1996); and two coelurosaurs, Santanaraptor placidus (Kellner, 1999) and Mirischia asymmetrica (Naish, Martill & Frey, 2004). The latter two likely had some feathery integument (Ji & Ji, 1996; Chen, Dong & Zhen, 1998; Ji et al., 2007). The first Mesozoic record of definitive avian dinosaur in Brazil, was only described recently, a fossil unearthed from the Crato Member of the Santana Formation (Carvalho et al., 2015a; Carvalho et al., 2015b). The feathers of Cratoavis cearensis (Carvalho et al., 2015b), bare interesting features, such as an extremely long rectrices, secondary remiges, alular feathers, and filamentous feathers. Regarding the remex and rectrices, there is no doubt that they were pennaceous feathers. Nevertheless, the filamentous elements may be a taphonomic artifact (Foth, 2012), these structures most likely were contour feathers or downy feathers. Patches with granulate spots rectrices may be associated with color patterns. However, no other evidence of its hue is given by the authors.

Evidence of feathers was not detected in any taxa of non-avian dinosaurs of the Araripe Basin (Kellner, 1999; Naish, Martill & Frey, 2004), event though feathers are considered plesiomorphic features for all taxa recorded here (Rauhut et al., 2012; Godefroit et al., 2013). This absence is odd given the vast record of soft tissues in both members (Crato and Romualdo), such as insect muscle fibers (Grimaldi & Engel, 2005; Barling et al., 2015), dinosaurs blood vessels (Kellner, 1996a), pterosaur wing membranes, muscle fibers, and headcrest (Martill & Unwin, 1989; Kellner, 1996b; Pinheiro et al., 2012), fish muscle tissue and stomach contents (Martill, 1989; Martill, 1990; Wilby & Martill, 1992), skin impressions of turtle (Fielding, Martill & Naish, 2005), fossilized microbodies related to pigmentation (Vinther et al., 2008), among others (cf. Martill, 1993; Martill, Bechly & Loveridge, 2007).

With the exception of Cratoavis cearensis (Carvalho et al., 2015b), feathered non-avian dinosaurs remain unknown in the Crato Member. Also, non-avian dinosaurs found in the Romualdo Member do not show evidence of feathers preserved with bones. Since their absence in both Lagerstätten (Crato and Romualdo) marks an unknown event, some possibilities emerge from it: (i) the non-avian dinosaurs of this deposits were glabrous (i.e., they did not possessed feathers) or were low in coverings; (ii) a selective taphonomic or geological process erased them; (iii) during the time of deposition, taphonomic conditions were very different between both members or even to the same unit, preventing their preservation; (iv) all possibilities may have happened simultaneously, or consecutively for the case of taphonomical and diagenetical processes; or, (v) these animals were not discovered yet. With the exception of the latter and the third, all other possibilities are regarded problematic. Firstly, the non-avian dinosaurs of the Araripe Basin were most likely covered with feathers or filament types, especially because these animals belonged to clades with feathered individuals (Ji & Ji, 1996; Chen, Dong & Zhen, 1998; Ji et al., 2007). Secondly, the process of fossilization in both units preserved tissues which are more prone to degradation (e.g., muscles fibers, blood veins), but in the Romualdo Formation it did not allow feathers that are relatively more resistant. It is important to state that analogous deposits with similar lithology (limestone rocks) and depositional settings, e.g., the Las Hoyas Formation in Spain (Sanz, Bonaparte & Lacasa, 1988; Sanz et al., 1996) and the Solnhofen Formation in Germany (Barthel, Swinburne & Morris, 1994), hold records of dinosaurs preserved similarly to the Araripe Basin. Especially in Solnhofen, feathers are present in the Archaeopteryx specimens, but they are not in the Compsognathus longipes Wagner, 1859 (Barthel, Swinburne & Morris, 1994) or Juravenator starki (Göhlich & Chiappe, 2006), suggesting that the third hypothesis may be true, once the selective taphonomic/ geological events can determinate the differential preservation of carcasses in the same depositional conditions. Especially to Santana fossils, the formation of concretions of the Romualdo Member may be responsible for obliterating these integumentary tissues since carcasses may have experienced some degree of transport and disarticulation, taken more time to be finally encapsulated. Differently, the Crato Member preserved rapidly and in situ, entire animals and climate may be responsible for the absence of non-avian dinosaurs, since in arid conditions are expected a low diversity of life (Stevens, 1989; Tilkens et al., 2007; Butler & Barrett, 2008). Therefore, it is possible to consider that non-avian dinosaurs may have reached the shorelines of the paleolake only occasionally, for food or water.

By their localization throughout the body, the feathers would be exposed to geochemical reactions during the initial phase of decay that followed the burial, being degraded early after exposing weathering or early diagenesis. However, as dinosaur remains were preserved within nodules, weathering is not responsible for the absence of these elements, since the dinosaur tissues remained relatively isolated from the surrounding environment throughout the geological time. It is expected that further studies may enlighten this odd absence.

To date, only a few records of feathers or filaments considered as ‘protofeathers’, were found associated with ornithischians dinosaurs (Mayr et al., 2002; Xu, Zheng & You, 2009; Zheng et al., 2009; Saveliev & Alifanov, 2014; Godefroit et al., 2014). Also, only skin impressions, osteoderms and ossicles of giant ornithischians and sauropods (including juveniles), were reported in the fossil record (Czerkas, 1992; Xu, Zhou & Prum, 2001; Coria & Chiappe, 2007; Christiansen & Tschopp, 2010; Arbour et al., 2014). In spite of the fact that true feathers were only reported in theropod dinosaurs, the poor record of feathers in ornithischia specimens and their absence in the entire subgroups (e.g., Thyreophora and Ornithopoda), as well as in sauropods, suggest that the preservation of these elements can be assigned to the sedimentological characteristics in which these animals were buried, representing a taphonomic artifact. In spite of occasional events of great sediment deposition, as well as the distribution of herbivorous dinosaurs over these sedimentary deposits (Butler & Barrett, 2008), according to extant examples (Behrensmeyer, 1978; Behrensmeyer, 1982), the preservation of their carcass (often huge) required more time to be completely buried.

The delay between death and final burial, might explain the absence of feathers alongside sauropods and great ornithischians bones. Furthermore, this slow process of preservation opposes the rapid burial suggested to the Crato Member (Martill & Davis, 2001). The sedimentary relationship has been examined in detail in dinosaur bearing deposits, such as the Morrison Formation and Judith River Formation of North America (Dodson et al., 1980; Wood, Thomas & Visser, 1988), but, to the Araripe Basin, they are based mainly in the fish fauna and restricted only to Romualdo (Martill, 1988; Martill, 1989).

Even though a recent study demonstrated that lithology itself may not be a sure factor for skin preservation of hadrosaurs (Davis, 2012), it may be an important factor, and perhaps decisive, factor in feather preservation. Another taphonomic feature that has to be considered is the type and grain size of the sediment that buried these animals (Barrett, Evans & Campione, 2015). Siliciclastic coarse grains, tend to preserve only larger hard parts of the animals (i.e., bones, keratinous beaks, tooths, and claws). Generally these sediments are related to high energetic depositional systems, with unidirectional flows, such as rivers and streams (Behrensmeyer, 1982; Behrensmeyer, 1988; Holz & Simões, 2002). Several evidences of Cretaceous enanthionithines and maniraptorans were found in sandstones of tidal, fluvial and flood plain deposits of Paraná Basin and São Luis-Grajaú Basin (Carvalho & Pedrão, 1998; Alvarenga & Nava, 2005; Novas, Ribeiro & Carvalho, 2005; Azevedo et al., 2007; Elias, Bertini & Medeiros, 2007; Machado, Campos & Kellner, 2008; Candeiro et al., 2012a; Candeiro et al., 2012b; Marsola et al., 2014; Tavares, Branco & Santucci, 2014; Delcourt & Grillo, 2015). However, with the exception of Cratoavis cearensis which was found in the carbonates of the lacustrine Crato Member of Araripe Basin (Carvalho et al., 2015a; Carvalho et al., 2015b), no other feathered dinosaurs were found in these deposits.

According to evidences in other non-avian dinosaurs (Mayr et al., 2002; Xu, Zheng & You, 2009; Zheng et al., 2009; Godefroit et al., 2014), it is possible that these elements were restricted to some regions of the body, in which they were not favored for preservation. We make here a parsimonious assignment of both feathers, GP/2E-7853 and GP/2E-7854, to the Coelurosauria clade. Because true pennaceous feathers were found in ornithomimosaurs (Zelenitsky et al., 2012), we assign GP/2E-8771 to the Maniraptoriformes clade. As pointed out by the large amount of evidence, both groups are responsible for these integuments in dinosaurs (Clarke, 2013).

Future perspectives

In a striped contour feather from the Araripe Basin described by Martill & Frey (1995), Vinther et al. (2008) have found oblate microbodies restricted only to the dark portions of the specimen. The light portions were markedly preserved as imprints. Those structures were previously interpreted as autolithified bacteria (Wuttke, 1983; Davis & Briggs, 1995), but subsequent studies revealed them as evidence of fossilized melanosomes (Zhang et al., 2010; Barden et al., 2014; Li et al., 2014; Egerton et al., 2015; Vinther, 2015; Lindgren et al., 2015a; Lindgren et al., 2015b). This interpretation enabled reconstructions of ancient color patterns of extinct animals, such as dinosaurs, birds, reptiles and fishes (Vinther et al., 2008; Vinther et al., 2010; Clarke et al., 2010; Carney et al., 2012; Field et al., 2013; Li et al., 2010; Li et al., 2012; Lindgren et al., 2012; Lindgren et al., 2014; Lindgren et al., 2015a; Lindgren et al., 2015b). As Vinther et al. (2008) demonstrated fossilized feathers from the Araripe Basin possess great potential in future taphonomical investigations, characterizing its importance in paleobiological studies of Mesozoic deposits.

Further investigations using Scanning Electron Microscopy equipped with Energy Dispersive X-ray Spectroscopy (SEM-EDS) will help in the identification of the presence of ultrastructures such as minerals, melanosomes, and other possible elements, confirming their preservation as described in the present paper. In addition, other techniques, such as Raman Spectroscopy (RAMAN), X-ray Fluorescence (XRF), Gas Chromatography-Mass Spectrometry (GC-MS), among others, also may add information such as its chemistry (Wogelius et al., 2011; Egerton et al., 2015), indicating possible taphonomic processes that occurred after deposition (Davis & Briggs, 1995; Schweitzer et al., 2008; McNamara, 2013).

Besides the study with ancient pigmentation, the application of these techniques are important, providing more information about these fossils, especially from the Araripe Basin, where fossil records of feathered dinosaurs is still limited (one enantiornithine and four non-avian theropods). These approaches not only allow a better understanding of the taphonomic and diagenetic processes that occurred in this basin, but enable future paleoenvironmental and paleoecological reconstructions (Li et al., 2010).

Conclusion

Despite the difficulty on the systematic approach, it is possible to identify isolated feathers of lower taxonomic rank, relying on the fossil record of the unit. For the Santana Formation of the Araripe Basin, the maximum taxonomic status reached is the Division (Coelurosauria) and Subdivision level (Maniraptoriformes). Based on the extinct and modern morphotypes, and on evolutionary models of feathers, the fossils were identified as two downy feathers (GP/2E-7853 and GP/2E-7854) and one semiplume (GP/2E-8771).

Although further geochemical analyses are being done, these feathers may be preserved as limonite (GP/2E-7853) and carbonized traces (GP/2E-7854 and GP/2E-8771); and the mechanisms which allowed the preservation of these elements were briefly discussed. As suggested by Martill & Davis (2001), it is also considered that these feathers have been transported into the paleolake by strong winds. Once in the waters, they sunk and were buried rapidly in the anoxic bottom. The absence of oxygen has an important role, once it prevented the activity of scavenging organisms, allowing its preservation. Nevertheless, other possible causes are also being considered, e.g., by predation (by fright molt).

The presence of avian and non-avian dinosaurs in the Araripe Basin is undeniable. Records of avian dinosaurs in the Crato Member consist of one bird and several isolated feathers, however evidence of non-avian dinosaurs remains unknown. On the other hand, in the Romualdo Member, four non-avian dinosaurs were described, but there are not yet formal descriptions of avian dinosaurs, nor even the presence of feathers associated directly with bones. However, in this unit, soft tissues were found in many animals including non-avian dinosaurs. Although unlikely, it is possible that a differential taphonomic process happened, preserving these non-resistant tissues instead of feathers.

Further geochemical investigations may reveal this process and how these specimens were preserved. Future investigations may also focus on the identification of the ultrastructures in addition to its chemical composition, offering the possible roles in life.

Despite their rareness and low taxonomic potential, fossilized feathers can offer insights about the paleobiology of its owners and the paleoecology of the Araripe Basin.

We are in debt to and deeply thankful for Jennifer Watling (University of Exeter and University of São Paulo) for the revision and suggestions made on the manuscript. We also would like to thank Luis Fábio Silveira (Ornithological Collection of the Zoology Museum—University of São Paulo) for the helpful comments and aid on the identification of these specimens. We also thank Ivone C. Gonzales (Institute of Geosciences of University of São Paulo) for the support in the accessing the Paleontological Collection and these specimens. And we are deeply thankful to Gabriel L. Osés (Institute of Geoscience of University of São Paulo), Bruno B. Kerber (Federal University of São Carlos—São Carlos), Cibele G. Voltani (São Paulo State University—Rio Claro), and Mírian L.A.F. Pacheco (Federal University of São Carlos—Sorocaba) for the support and invaluable comments. We thank T. Alexander Dececchi, Natasha Vitek and Mark T. Young for the helpful and invaluable comments on the manuscript.

Additional Information and Declarations

Competing Interests

Author Contributions

Data Availability

The authors declare there are no competing interests.

Gustavo M.E.M. Prado conceived and designed the experiments, performed the experiments, analyzed the data, contributed reagents/materials/analysis tools, wrote the paper, prepared figures and/or tables, reviewed drafts of the paper.

Luiz Eduardo Anelli conceived and designed the experiments, analyzed the data, contributed reagents/materials/analysis tools.

Setembrino Petri wrote the paper, reviewed drafts of the paper.

Guilherme Raffaeli Romero analyzed the data, contributed reagents/materials/analysis tools, wrote the paper, prepared figures and/or tables, reviewed drafts of the paper.

The following information was supplied regarding data availability:

The research in this article did not generate any raw data.

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
