# Peer review of "New occurrences of fossilized feathers: systematics and taphonomy of the Santana Formation of the Araripe Basin (Cretaceous), NE, Brazil"

_PeerJ, doi:10.7717/peerj.1916_

## Round 0.1 · original submission · Major Revisions

Dear Authors,

Your manuscript has been reviewed, and I have accepted the decision of 'major revision' from the reviewers.

I have two comments (in addition to those of the reviewers) that the authors should address prior to resubmission:

1. Authority and date should be provided for each species-level taxon at first mention. Please double-check that the nominal authority is also included in the reference list.
2. Please write lithostratigraphic units out in full (i.e. please do not abbreviate Formation, etc).

Once again, thank you for submitting your manuscript to PeerJ, and I look forward to receiving your revision.

·

Basic reporting

This manuscript is difficult, at times impossible, to understand because of problems with the English. The abstract has been edited to give an example of the degree of improvement necessary, but the entire manuscript contains similar errors that should be fixed to make the manuscript more understandable.

Otherwise, the introduction provides an appropriate level of background information and helps to justify the use of a taxonomy and morphotype classification scheme for the feathers described later in the manuscript.

Moving lines 306-310 to the Geologic Setting section would improve clarity and make it easier for the reader to find information about the provenance of the fossils.

Figure 3 can be removed without loss of information. I agree with the authors that it is important to know that feathers more complex than filaments have only been reported for species within Coelurosauria, but that information and the phylogenetic tree that supports it was already reported in Xu et al. 2014 (cited by the authors) in more detail than is provided here. It would be sufficient for the authors to cite Xu et al 2014 to support their identification of the fossilized feathers as belonging to Coelurosauria. It is unnecessary for them to create a new version of a figure that already exists.

It does not seem necessary to give each feather identified as Coelurosauria incertae sedis a separate section. A lot of information about horizon, lithology, age, taxonomy, etc. ends up repeated. The description of those two feathers could be combined into a single section.

The review of potential origins of the fauna (391-431) is not relevant to the new findings presented in this manuscript and can be deleted. The origin of the fauna, whether allochthonous or autochthonous, does not change the identification of the feathers, their potential positions on the body and functions, or their preservation.

Experimental design

The materials and methods section had an appropriate level of detail. It looks like there are some interpretive dotted white lines in figure 2E that might help the reader understand what was identified and measured. Please label the white dotted line to help the reader.

Validity of the findings

Evidence is provided in the written description to justify the identification of each feather. The manuscript would be improved if the barbules and could be figured and labelled, as was done for the calamus in Figure 2E-F to strengthen the description and identification of the feathers.

Also, are authors sure that the structure labelled BI in Figure 2F is a barbule and not a barb? It looks more like the size of a barb than a barbule to me. If it is a barbule, please provide evidence in the text to explain.

The interpretation of the cause of the lack of reported feathers associated with non-avian dinosaurs appears contradictory. The authors claim that the possibility of differential preservation within the same unit is problematic (501-502), then go on to develop the possibility as the most likely scenario of the six presented (505-514). The authors should clarify their arguments. It may be that simple improvement of the English will clarify this apparent contradiction.

I do not see the justification for including the word “paleoecology” in the title. No new information about the life habits of any animal or the interactions between different animals is provided in the paper. Otherwise, the title is justified by the evidence provided and the interpretation of that evidence.

Additional comments

The scientific advance of this manuscript, besides the reporting of new fossils from a Brazilian lagerstatte deposit, is to interpret as much as possible from isolated feathers by applying models of feather evolution and our current understanding of feather morphology and the phylogenetic context of feather evolution. Using that context, the authors identify the feathers as representatives of the clades Coelurosauria and Maniraptoriformes, which is consistent with previous reports of bones belonging to members of those clades. The authors also use details of the fossil preservation to hypothesize about the chemical preservation type of each fossil. The contribution of the manuscript is highlighted well by the authors in lines 294-296.

·

Basic reporting

No comments

Experimental design

No comments

Validity of the findings

While I believe the validity of the findings in general, on occasions the authors extend beyond their data. These instances are noted in the comments to authors and do not significantly reduce the validity or impact of the manuscript.

Additional comments

I enjoyed reading this paper very much, but I do have some concerns. Though the science and the finding is sound, the writing makes it difficult through. I fear the sentence structure and grammar issues reduce the impact of the results significantly and they need to be addressed before moving this manuscript forward. Also there are large sections that are either unnecessary due to repetition or because they contain information not relevant or informative to the study at hand. I believe eliminating or minimizing them would both make the manuscript significantly more concise and enhance its impact.

Beyond the sentence structure, some of which I have tried to give my suggestions to correct below, I do have some more significant issues that should be address. Despite this I believe that this paper is a significant contribution, as it tries to bring isolated feathers into the systematic light. I look forward to reading further versions of it.

Significant issues

Line 77: What does precocity have to do with feather morphotypes? It is likely that many non-avian theropods had precocial young see Weishampel et al. 2008 “New oviraptorid embryos from Bugin-Tsav, Nemegt Formation (upper Cretaceous), Mongolia, with insights into their habitat and growth”

Line 87: Given the multiple methodological issues raised regarding this study in the responses, I would be cautious in this conclusion based on this study.

Line 91: This is unlikely; multiple studies suggest that endothermy evolved earlier, not later, in the Ornithodiran lineage

Line 197: Since all are in table 2 I do not believe this section is needed.

Lines 267-290: Should this part not be assimilated into the individual descriptions?

Lines 296-298: I would suggest that there is a limit to the taxonomic granularity that can be achieved, since downy feathers can be juvenile features of derived stem (or even true) avians or found in adults of more basal forms. Do you have any evidence that it is due to the lack of specimens (that does not hamper other systematic works)?

Lines 325-335: Where did this conclusion come from and on what basis? You stated these were mophotype IIIb, that they were "downy" but never gave justification for this assignment. Besides placement on the organisms what morphological characters distinguish these feathers. Please list them and why these specimens conform more closely to them than to feathers on other locations. This significantly influences other of the inferences you draw in this section.

Lines 338-340: These are statements are speculative since you do not know the owner, the colour or the position on the body.

Lines 356-360: These line are unnecessary and can be eliminated.

Lines 398-400: Please give evidence for this assertion.

Lines 400-431: Much of this section is not necessary for this paper. Please trim significantly. Also it should be in the introduction when you set the stage of the locality, not in the discussion.

Lines 557-558: Sinosauropteryx and Sinocalliopteryx are basal coelurosaur, though in some analyses its a basal maniraptorans, not basal theropods. Sciurumimus plylogenetic placement is debated, many think it is actually a coleurosaurs and Yutyrannus is a tyrannosaur. None of these are basal theropods.

Lines 567-591: This entire section is not necessary, if you wish to retain it I suggest you rewrite it significantly to integrate it more into the paper as a whole.

Minor changes

Line 54: Change to "Recent molecular studies"

Lines 58-60: This is a bit broad of a statement, the tool kit and the way of turning those tools into a final product are different enough. Bird feathers are different from amphibian integuments. Also the idea that integument filaments are now consider symplesiomorphic for tetrapods is not what Lowe et al. 2015 show. They have all the patterning genes coming in by Amniota, but the CNEE regulatory elements are not all in by even Archosauria.

Line 64: It was obviously too late for you to have added this in the first draft but please include Barrett et al. 2015 "Evolution of dinosaur epidermal structures" in the next version.

Line 66: Saurischians denote an unranked clade of dinosaurs, I think you mean either archosaurs or saurian

Line 67: Maybe use Ornithodira instead of Archosaurs, as the later is too broad.

Line 73: Even non "modern" morphotypes are still feathers, perhaps change wording to non-filamentous feathers or derived feathers.

Line 84: All these ornithischians are not the best example, try including things like Ornithomimids (Zelenitsky et al. 2012) or more basal theropods.


Line 95: Except for a single ornithischian taxa (Varrichio et al. 2007), I am unaware of any other fossorial dinosaurs. Thus I would avoid associating sensory tactile protofeathers with this life history strategy. As it is the text you cite only mention this in passing using the example of the derived maniraptoran Alvarezsaurids, which likely postdated the origin of your "modern" feathers.

Line 96: This last part of the sentence is confusing, please re-word. I believe you meant to say that this proposal currently lacks evidence.
Line 97: Add the word “another” before “the possible reason….”

Lines 97-99: This sentence is very wordy and can be condensed to make it both clearer and more concise.

Line 103: With some editing I believe you can condense all the three major hypotheses for feather origins stated here into one paragraph that reviews the thoughts on the origin of feathers.

Lines 103-104: Aren't feathers, by definition soft tissue? Do you mean other soft tissue preservation, if so please give an example.

Lines 109-110: It is not immediately clear, and needs to be for the next sentence that you are talking about feathers only, not konservat-lagerstatten as a whole. Maybe instead of vestiges state simple that only a few deposits preserve feathers and/or other ornithodiran filamentous integument.

Lines 111-119: Instead of listing so many references that overlap for some of these localities can you not condense these into larger groups (like Europe and North America) and then use Kellner 2002 supplemented when necessary by a couple of other papers? As it is you have 24 references listed here, I feel that is too many.

Lines 120-122: Please re-word, perhaps condense into a single thought and eliminate unnecessary verbiage.

Lines 130-131: Eliminate the period, as well as “here we”

Line 137: Place "situated between 38. 30' to 40. 50' W of longitude, and 7. 05' to 7. 50' S of latitude (Coimbra et al., 2002; Vianna & Neumann, 2002)" after Brazil.

Lines 143: Replace “that depends on authors approaches” with “differing interpretations "

Lines 145-147: Reword to something like " We adopt Assines (refs) descriptions since these works cover the entire basin in great detail and are in accordance with the Brazilian Code of Stratigraphy."

Line 155: eliminate “with”

Lines 155-156: Please reword since it is not clear.

Line 192: What morphotype is it? Looks like a Prum stage IIIb. Please be clearer on what dimensions you are referring to eliminate it from being a contour or pennaceous feather.

Line 200: Do you mean that the slight weathering is producing these minerals? Also what does “etc” signify, are you just trying to list the weathering products?

Lines 221-222: Remove “It is possible to notice that”.

Line 243-244: Please reword to something like “and the largest of the three”

Line 247: Perhaps reword to "hooklets are not preserved"

Line 253: remove “an”

Lines 263-265: But you don't assign it to Aves, also do not some paravians have extremely similar feathers to this?

Line 300: You stated this is the first taxonomic approach applied to isolated feathers did you not? Do you mean by "improper" an incorrect approach given our new level of knowledge of feather development and evolution?

Lines 308-309: remove from “many characteristics…..this unit” and change plausible to possible.

Line 312: all 3 are from the Mesozoic

Line 313: please change to “ontogenetically early and evolutionarily basal”

Lines 318-322: These sentences should be reworded to make them clearer.

Lines 341-344: These lines should not this be in the next section, given you discus taphonomy.

Lines 444-445: Please remove this sentence.

Line 447: The fright mold sentence is unnecessary.

Lines 452-457: Are these lines needed? This information is interesting but does not inform this paper.

Lines 458-460: Not needed

Lines 461-468: Please reword, I believe you meant that the later two species are classified in clades that posses feathers and thus are reconstructed as having them as well.

Line 478: add definitive

Lines 486-489: Not necessary

Lines 490-494: please rewrite, it is not clear

Lines 503-505: Please remove this sentence.

Lines 526-531: Please reword, as the main point (size factors preventing rapid burials thus allowing feather degradation, is not very clear on a first reading.

Lines 532-540: An edited version of this paragraph, perhaps excluding the Ediacara fauna should be merged with the proceeding one to give two potential reasons for lack of feathery integument in these dinosaurs.

Lines 542: Please remove the second part of this sentence, it is unnecessary after the first

Lines 545-551: The previous section justifies your focus on theropods,this section is not needed.

Line 553: or even at the family level

---

## Round 0.2 · Minor Revisions

Dear authors,

Your manuscript has been re-reviewed, and I have accepted the decision of 'minor revision' from the reviewers. Both reviewers have highlighted language issues which still need to be addressed. Please see the comments of reviewer two in particular.

Once again, thank you for submitting your manuscript to PeerJ and I look forward to receiving your revision.

·

Basic reporting

The English throughout the manuscript could still be improved for clarity and ease of understanding, but otherwise submitted article is adequate in terms of structure and and scope.

Experimental design

No Comments

Validity of the findings

No Comments

·

Basic reporting

No Comments

Experimental design

No Comments

Validity of the findings

No Comments

Additional comments

This is a very well done paper and I believe strongly that it should be published. Still I have a few suggested revisions, mostly based on rewording sentences to improve clarity and eliminating redundant or more speculative sections. These are all minor issues that should not prevent or delay the publication of the manuscript.

Line 65: Does not the Barret et al. 2015 paper you cite challenge this? While I agree that Godefroit et al. 2014 needs to be referenced in this section, I think it fits better above, with Xu 2006 and Norell 2001. I think that starting this paragraph with “However” is the best course of action, as the however works well in linking the thought on “modern type featehrs” with the taphonomic issues raised in the next sentence.

Line 75: Again I must point out that precociality and feathers are not linked. Embryonic therizinosaurs, which are not suspected to have large pennaceous feathers are suspected to by highly precocial (Kundrat et al. 2008). Even more basal theropods such as Coelophysis (Jasinski 2011), Allosaurus and Tyrannosaurs (Lee and Werning 2008) are suggested to have been precocial. Thus precociality may be a plesiomorphic trait for coleurosaurs, if not theropods in general. Yes basal birds look to have been percocial, but that likely had nothing to do with feather devolpment and thus I don’t agree that it needs to be mentioned here.

Line 81: Sligtly awkward wording, but I get the point of the phylogenetic disconnect between feather origins and flight origins. My I suggest a slight rewording to something along the lines of “, but recent findings of non-volant yet feathered coleurosaurs (your refs) as well hypothezied pre-flight functional hypothsis for feathers origins and evolution (Dial refs) clearly show a disconnect between feather and flight”.

Line 90: There is much discussion on this since the oldest evidence of pycnofibres in pterosaurs is Mid to Late Jurassic and if you trust Andres et al. 2010 phylogeny the oldest lineage which we can reliably link them too is Upper Jurassic, 10’s of millions of years after pterosaurs origins. Also the purported sister to pterosaurs, Scleromocholus, is reported to have scales, not fibers (Benton 1999). Thus I would say something like “thermoregualtry coverings (pycnofibers and featehrs) whether analogous or homologous are suggested to be an ornithodiran wide triat, but this his highly contenious”.

Line 129: Suggested rewording to something like “Although the Romualdo Member during this time was characterized by the maximum marine transgression, where the salty waters entered this basin with N-NW direction (Assine 1994).”

Line 134: Suggested rewording to something like “Nonetheless, the absence of ichnofossils in the rocks of this unit require further investigations.”

Line 137-153: I like the discussion on possible other causes for preservation of terrestrial vertebrates in these beds, but I believe they take too much space. Please shorten them to a single paragraph combined as this is not the focus of the paper.

Line 175-177: maybe reword slightly.

Line 183: add “, and” after age

Line 306-308: please reword

Lie 352: Not 100% convinced. If these feathers had a different size ratio in non-avian theropods (or in stages of their ontogeny) they could easily be from something larger than this.

Line 387: remove “albeit meager”

Line 391: remove “allowing the presence”

Line 414: Please reword. I think the point is that avians may have lived by the shore, non-avian dinosaurs inland, but this is also slightly against the sentence above it and is speculative. Could not feathers simply be better able to drift out (either on the wind or water) into the lake where they could fossilize while whole organisms were harder to preserve?

Line 4423: remove: ”once it is”

Line 425: replace “Therefore the” with “though this”

Line 436: reword slightly to something like the following “During their life, birds tend to lose feathers by a variety of means such as through ontogenetic or seasonal molts, and under high stress situations.”


Lines 439-445: Please reword or shortened to something like “ Isolated feathers can also be the result of predation by aquatic predators but we have no direct evidence of it yet.”


Lines 447-449: Please reword to something like “The carbonate concretions of the Romualdo Member of the Santana Formation, provided a record of at least four non-avian theropods, with the possibility of a fifth (Machado and Kellner (2007)”

Line 452: Perhaps say, “the latter two likely had some feathery integument” then give the references.

Line 462: The thought “It is speculated, that filament feathers were present even in megalosauroid dinosaurs (Rauhut et al., 2012), but according to previous reports, …” should be changed to something like “Evidences of feathers were not detected in any taxa of non-avian dinosaurs of the Araripe Basin (Kellner, 1999; Naish et al., 2004) though feathers are deemed to be plesiomorphic feature for all taxa recorded here (Rauhut et al. 2012, Goedfroit et al. 2013).”

Line 478-480: A repeat of last paragraph and can be removed

Line 500: What about option 1? A quick sentence dismissing it would be nice.

Line 509 : And why would that make it harder for you to find feathers on the ones you do find?

Line 515: This sentence could use some referential support. I know you are speculating, but is there any evidence that the process you suggest can occur in te way you propose?

Line 532: Do you mean Davis 2012?

Lines 517-541: I understand what you are trying to say, but this does not impact this paper and is speculative. Another explination is that they lost most if not all of their “feather” integument. We don’t actually know why these groups don't tend to be fuzzy. Again this does not impact your paper and I think it should be excluded.

---

## Round 0.3 · Minor Revisions

Dear authors,

I have looked over your manuscript, and I am satisfied with your response to the reviewers comments. However, as the reviewers have previously noted, there are some spelling and grammatical issues. Looking through your manuscript some of these are still present.

Some grammatical tips:
1. Try not to use one sentence paragraphs (e.g. your final paragraph in the conclusions)
2. Try not to start sentences with 'and' or 'because'.

Also, in the acknowledgments, my first name is Mark not Mike. Can you have an English speaking colleague give your manuscript a read through, looking at the language? Once done I can see no reason not to accept your manuscript.

Once again, thank you for submitting your manuscript to PeerJ and I look forward to receiving your revision.

---

## Round 0.4 · Minor Revisions

Dear authors,

After the previous rounds of review the science in your manuscript is considered accepted. However, the language still requires some work. This is as much for your benefit, as it will make your final paper more readable and valuable.

Please find below my edited version of your abstract as a starting point, but please also carefully review the full manuscript once more for English language, as PeerJ does not offer copyediting

Here we describe three fossil feathers from the Early Cretaceous Santana Formation of the Araripe Basin, Brazil. Feathers are the most complex multiform vertebrate integuments; they perform different functions, occurring in both avian and non-avian dinosaurs. Despite their rarity, fossil feathers have been found across the world. Most of the Brazilian feather fossil record comes from the Santana Formation. This formation is composed of two members: Crato (lake) and Romualdo (lagoon); both of which are predominantly reduced deposits, precluding bottom dwelling organisms, resulting in exceptional preservation of the fossils. Despite arid and hot conditions during the Cretaceous, life teemed in the adjacency of this paleolake. Feathered non-avian dinosaurs have not yet been described from the Crato Member, even though there are suggestions of their presence in nearby basins. Our description of the three feathers from the Crato laminated limestone reveals that, despite the small sample size, they can be referred to coelurosaurian theropods. Moreover, based on comparisons with extant feather morphotypes they can be identified as one contour feather and two downy feathers. Despite their rareness and low taxonomic potential, fossilized feathers can offer insights about the paleobiology of its owners and the paleoecology of the Araripe Basin.

---

## Round 0.5 · accepted · Accept

Dear authors,

Thank you for your quick response. If you are now happy with the language used in your manuscript I will accept it for publication in PeerJ.

---

## Author Rebuttal · Round 0.5

Gustavo M. E. M. Prado
Bloco C (Térreo) - Sala 3
Rua do Lago, 562 - Butantã.
CEP: 05508-080 São Paulo - SP, Brazil
gustavo.marcondes.prado@usp.br

March 16th, 2016

To Mark T. Young
Academic Editor of PeerJ
PeerJ, Inc.
PO Box 614
Corte Madera CA 94976 USA

Dear academic editor,

According to your suggestion, which highlighted the problems in English language, we sent the manuscript to be reviewed by a British fellow researcher. Thus, the uploaded manuscript is already the revised and final version. We hope that now, all needed changes were fulfilled in accordance to PeerJ standards.

Sincerely,

Gustavo M. E. M. Prado
Guilherme Raffaeli Romero
Luiz Eduardo Anelli
Setembrino Petri
Institute of Geosciences - University of São Paulo